# Exploration of the Evaluation and Optimization of Community Epidemic Prevention in Wuhan Based on a DEA Model

**DOI:** 10.3390/ijerph17207633

**Published:** 2020-10-20

**Authors:** Yuxiao Zhang, Peiyu Cao, Jiejie Meng, Jiuyun Qiu, Qiwen Hu, Lei Cheng

**Affiliations:** 1Global Health Institute/School of Health Sciences, Wuhan University, Wuhan 430071, China; 2School of Health Sciences, Wuhan University, Wuhan 430071, China; 2017302170001@whu.edu.cn (P.C.); mengjiejie@whu.edu.cn (J.M.); qiuJiuyun@whu.edu.cn (J.Q.); huqiwen@whu.edu.cn (Q.H.); chenglei-cherry@whu.edu.cn (L.C.)

**Keywords:** community, epidemic, DEA, co-word analysis

## Abstract

Background—Communities played a key role in preventing the spread of coronavirus, not only during the threshold period of the epidemic but also in the normal stage of prevention. Scientifically evaluating the community’s work is necessary for prevention in the normal period of the epidemic and can provide a reference for the management of different countries. Methods—Based on data envelopment analysis (DEA), this article used community worker data to evaluate the matching of service supply and demand during the epidemic period and used co-word analysis to analyze the content and the residents’ demands for community service from the threshold period to the normal period of the epidemic. Results—According to the results of the DEA model, early in the epidemic, 13 of the 15 districts’ DEA values were invalid, indicating that there was a shortage in community workers in Wuhan. The results of public opinion analysis showed that from the threshold to the normal period of the epidemic, the emphasis on community service gradually transformed from epidemic prevention to an integrated service, which effectively met the composite service needs of community residents for both prevention and life. Conclusions—In the face of public health emergencies, the government should ensure an adequate number of service personnel, mobilize the service resources, refine the service content, and adjust the incentive policy, which can help to improve the quality of residents’ lives and the coordination degree of the prevention and control as part of the epidemic control in the emergency period and the social and economic recovery after the epidemic.

## 1. Introduction

The community is the basic unit of society. In the sense of human resources science, the community is formed by a group of connected people living and working in a specific area [1]. Usually, communities are not administratively divided, but in China, they are included in the administrative system and managed by the government. The community staff in China are generally made up of civic workers made up of management, service workers, and security officers [2]. In recent years, some cities in China have set about exploring the PPP (Public-Private Partnership) model. During the coronavirus epidemic, the government and the market have shown their increased willingness to cooperate. Yan Oi Tong, a social welfare institution in Hong Kong, is partially sponsored by the Social Welfare Department and along with commercial fees, these funds are used to provide community services. The institution recruits and trains employees in their own mold. Some developed countries also use similar management. Communities in Germany are generally managed by property companies and most of the infrastructure, such as the kindergartens and the hospitals, are privately managed.

Nowadays in China, the communities adopt a grid management model. The model divides the residential area into smaller service units in terms of the buildings and business companies, which lays a solid foundation for the intellectualization of community governance. However, the current division mode overfocuses on the physical environment and neglects the internal and external contacts of the community, along with accurate targets based on the needs of community residents [3]. Therefore, the communities fail to provide services according to the demands of the residents and are unable to play a constructive role in grid management [4,5].

The community was a key and special part during the epidemic in Wuhan. It took on the responsibility to improve the public health environment and to help with the registration, transportation, and treatment of the patients in the community. In addition, it also provided the basic necessities of life for the residents during the middle and later stages of the epidemic. The community not only acted as the main organization of the public health emergency system but also committed to urban governance (see Figure 1). The coronavirus disease 2019 (COVID-19) epidemic is a whole new challenge for humanity. Medical service in cities, especially in metropolises, is under great pressure. At the beginning of the epidemic, communities were trapped by such problems as resource shortages, a lack of service, and a poor information platform, which had caused difficulties in the control of the spread of the epidemic. The headquarters took a series of measures in time to supply service personnel, optimize the service model, and provide a variety of community services. Thanks to these timely measures, the new confirmed cases per day began to decrease and the risk of the disease spreading kept declining by setting up a multi-sectoral cooperation platform, swiftly scaling up epidemic emergency capacity, whole-of-society actions with engagement of social organizations, and engaging citizens in the epidemic prevention and control. To optimize the supply of community service resources and to promote the community service capacity in regular epidemic prevention and control, it is necessary to analyze the supply and demand gap of community services during an epidemic. Since community services are quite complex because of the diversity of the personnel composition and the multiple levels of the service demand, it is time to introduce a more efficient analysis model, which is named the data envelopment analysis (DEA) model. The DEA model is one of the most commonly used efficiency evaluation methods for a DMU (decision-making unit) to study multiple inputs and outputs [6]. This method has been widely used in the evaluation of commercial service performance and the allocation of public service personnel due to its advantages [7]. It specializes in calculating the relative efficiencies of multi-input and multi-output DMUs without the distraction of dimensional unification. As the basic model in DEA, CCR (A. Charnes & W. W. Cooper & E. Rhodes) can not only measure the efficiency of its evaluation object but also evaluate its technical and integrated efficiency, while obtaining the slack variables and its scale income. Therefore, CCR can better describe resource allocation, such as an actual gap and a change of human resources in community services during the coronavirus epidemic.

Big data analysis of public opinion is a means to track the opinions and demands of network users under specific circumstances, which has been widely used in public management, marketing, and other disciplines [8]. Co-word analysis is a common method of bibliometrics and is a kind of content analysis. Its basic principle is to count the occurrences of a group of keywords in the text and reflect the strength of correlations between these keywords through a series of analytical work. Finally, it shows the core content of the text [9]. Its advantage is that it can collect real-time content of Internet information, 24 h a day. The obtained information can be fully retrieved and automatically updated. It can be used to analyze the evolution of public opinion themes and the analysis of time trends, especially for the complex needs of people’s livelihoods under emergencies. Social network analysis (SNA) is a commonly used quantitative analysis method in sociology and reduces unreasonable clusterings. Therefore, constructing a co-occurrence network knowledge map for the relationship between the high-frequency keywords of service types and service demands can make the research results more credible [10]. With 904 million netizens in China, the Internet has become an important place for people to express their opinions. During the epidemic, the Internet has become the main place for people to talk about their needs and to seek help [11]. The co-keyword analysis of network text is one of the commonly used methods used to analyze service content and demands, which can reflect the changes in community service content and demand more intuitively through the co-occurrence analysis of high-frequency keywords.

This study defined the representative stages of the epidemic situation according to the changes in epidemic prevention and the control situation in Wuhan, and used public opinion analysis to sort out the actual work items in terms of disease prevention, control, and life services, and to determine the subject of service.

From the perspective of service supply, the changes in service gaps in each stage were objectively assessed based on data envelopment analysis.

From a service demand perspective, residents’ needs and expectations were analyzed based on big data public opinion analysis, and the change in community service attention content from the residents’ perspectives is presented.

Wuhan has now entered the normal period of epidemic prevention and control. The city now turns its attention back to socio-economic development and the basic infection prevention measures. Communities will play an important role. This study scientifically evaluated the previous model of epidemic prevention and control work in the community to provide experience for the normal period of prevention and control to offer a basis for new work concerns and to give a reference for management models in different countries.

## 2. Method

### 2.1. DEA Model

#### 2.1.1. CCR-DEA Model

Based on the concept of “relative efficiency,” the DEA model is a systematic analysis method that is used to evaluate the relative effectiveness or benefit of a DMU according to multi-index inputs and outputs by using convex analysis and mathematical linear programming theory. It can measure and evaluate the relative effectiveness of the DMU with multiple inputs and outputs [12]. CCR is one of the classical efficiency evaluation models in the theoretical system of a DEA model. Its principle is to assume that the DMU has a fixed scale income to measure the total efficiency.

##### Variable Definition

Assuming there is a set of DWU*_s_*, {DWU*_j_*: *j* = 1, 2, ⋯, *n*}, each DWU*_j_* consumes multiple positive inputs *x_ij_* (*i* = 1,2, ⋯, *s*) to produce multiple positive outputs *y_rj_* (*r* = 1, 2, ⋯, *m*), *x_ij_* represents the total input of the *j*th DMU to the *i*th input, *y_rj_* represents the total output of the *j*th DMU to the *i*th of output. *V_i_* refers to the weight of the *i*th input and *u_r_* is the weight of the *r*th output.

##### Efficiency Evaluation Model

Corresponding to the weight coefficients *v* = (*v*_1_, *v*_2_,…, *v_s_*) and *u* = (*u*_1_, *u*_2_,…, *u_m_*), each DMU has a corresponding efficiency index. Taking the efficiency index of the *j_0_*th DMU as the goal and the efficiency index of all the DMUs as the constraint, dual planning is obtained by constructing the CCR model by using the Charnes–Cooper transformation, introducing duality theory, and adding slack and surplus variables.
$$s.t.\{\begin{array}{c} \min\theta ,\\ \sum_{j=1}^{n}\lambda_{j}y_{j}-s^{-}=y_{0},\\ \sum_{j=1}^{n}\lambda_{j}x_{j}+s^{+}=\theta x_{0},\\ \begin{array}{l} \lambda_{j}\geq 0,j=1,2,\cdots ,n,\\ ,n,\\ s^{+}\geq 0,s^{-}\leq 0. \end{array} \end{array}$$

##### Economic Significance of the CCR Model

If the optimal solutions are *θ**, *s**^+,^
*s**^−^, some conclusions for the validity of the CCR model are obtained as follows:①If *θ** = 1, *s**^+^ = 0, and *s**^−^ = 0, then DMU*_j0_* displays DEA efficiency. This means that the inputs and outputs of each DMU reach the optimal state.②If *θ** = 1 and at least one of *s**^+^ or *s**^−^ is greater than zero, the DMU*_j0_* displays weak efficiency.③If *θ** ≠ 1, the DMU*_j0_* shows no efficiency. The inputs and outputs of each DMU do not reach the proper ratio.

##### Evaluation of the return of scale:

①When ∑*λ_j_** = 1 (*j* = 1, 2,…, *n*), the scale return remains unchanged, which means the production has reached a relative optimum.②When ∑*λ_j_** < 1 (*j* = 1, 2,…, *n*), the scale return is increasing, which means that increasing the input of a DMU by one unit can bring more output than a unit.③When ∑*λ_j_** > 1 (*j* = 1, 2,…, *n*), the scale return is decreasing, which means that one unit of input of each DMU will only get less than a unit of output.

#### 2.1.2. BCC-DEA Model

The CCR model assumes that the marginal benefit of a DMU is constant, but in practice, the marginal benefit of a DMU is often dynamic. Adding a hypothesis based on the CCR model, the BCC model can calculate the true technical and scale efficiencies, which makes up for the deficiency of the CCR model.

#### 2.1.3. Data Sources

The data of community residents and new confirmed cases of COVID-19 pneumonia came from community websites and Wuhan epidemic reports, respectively. The number of the community service personnel came from network reports and community service online platforms (such as Wuhan Microneighborhood), and was supplemented by semi-structured interviews to determine and adjust the actual number of community service personnel.

Based on a DEA of public services, this study conducted stratified random sampling according to the community density to select typical communities and used semi-structured interviews with community leaders to obtain the data regarding the community workers, volunteers, and government staff applying to participate in community work during the epidemic. Furthermore, this study analyzed the frequency of people serving the community and the geospatial distribution of community in Wuhan based on community human resources data in 1406 communities of 15 districts in Wuhan, and analyzed the resource allocation of Wuhan community services by using the DEA CCR model and Deap 2.1 software (UNE, Armidale, Australia) source to evaluate the actual gap and the changes of actual service personnel deployment in the Wuhan community during the epidemic.

### 2.2. Public Opinion Analysis

The public opinion analysis included three links: constructing the database, gathering the text, and analyzing the text. First of all, we used the well-known public numbers in Wuhan from web pages such as Changjiang Daily and Wuhan Release as the monitoring data source. The tweets and comments from 23 January 2020 to 20 May 2020 were monitored, and the data were captured and aggregated using a web crawler. High-frequency keyword clustering analysis involves the analysis of the relationship between high-frequency keywords in the text, which reflects the similarity and heterogeneity of high-frequency keywords. Through mapping these relationships, we could further excavate deep information, such as text content and social semantics, using semantic analysis to identify hot topics related to community services during the epidemic period and collect and organize keywords, such as community management and community prevention. According to the changes in the number of new confirmed cases of COVID-19 pneumonia, the prevention and control of the epidemic situation was divided into three stages (the threshold period was from 23 January to 5 March, the low-incidence period was from 5 March to 25 April, and the normal period was from 26 April to 20 May). Using Excel statistics of the co-occurrence of high-frequency keywords in the community in official information and residents’ opinions, netdraw software was used to draw a social network map, connecting each high-frequency word through SNA to form a high-frequency word network. The node size represents the word frequency, the location of the node represents the center degree, and the relative number of the lines represents the correlations between the words, which directly reflect the main expression logic and concerns of the network text, horizontal and vertical analysis of the prevention and control period to the normal period of community work content, and the residents’ service demand transformation trends.

### 2.3. Ethical Statements

The study protocol was reviewed and approved by the Institutional Review Board of Wuhan University (IRB number: J-17-2016) within China.

## 3. Results

### 3.1. Analysis of the Community Service Content and Quantity at Different Stages

#### 3.1.1. Community Services at Different Epidemic Stages

The results of the co-word analysis showed that the main service content of the community was constantly changing in different stages of the epidemic to meet the needs of the community residents (see Figure 2, Figure 3 and Figure 4). In the threshold period of the epidemic, community services mainly focused on anti-epidemic goals, including disease monitoring, buying drugs for chronic patients, and daily supplies’ purchasing and distribution. In the low-incidence period of the epidemic, the main services provided by the community were gradually enriched and were not limited to epidemic prevention and the purchase and distribution of supplies, and the services for special groups were gradually increasing. After the epidemic entered the normal period, the main community services gradually moved toward integrated services.

#### 3.1.2. Deployment of Community Human Resources in Different Stages

In the normal period, the main structure of the community staff in Wuhan included grid managers (service personnel who took the residential building as the unit), administrative personnel, life service personnel, and security personnel. During the epidemic, the need for community services and the treatment of diseases increased, while community personnel were in short supply. To estimate the human resource gap of the community, we selected the following indicators for the model calculations (see Table 1):

In the model calculations, there was one input indicator:

Number of community workers, including grid managers, administrative personnel, life service personnel, and security personnel.

There were two output indicators:

Number of community residents, which refers to the output of community work in life services.

Number of confirmed patients in the community, which refers to the output of the community work regarding sanitation services.

Semi-structured interviews were used to obtain the input data of the community during the epidemic. The analysis of the service gap and its change in the community before and after the implementation of community health policy involved calculations using the DEA model, where the results are presented as follows (see Table 2).

The results showed that in the early stage of the epidemic, only Wuchang District and Qiaokou District had an efficiency value of 1, that is, the DEA was effective, while the other 13 districts were in a state of irs, which means that the return to scale was increasing. Human resources of most of the communities in Wuhan failed to meet the needs of disease prevention and the control and life services during the epidemic. Increased investment in the human resources of communities in Wuhan can enhance the community service capacity.

#### 3.1.3. Community Service Content of Wuhan Residents’ Concerns against the Background of Novel Coronavirus as Reflected by Network Public Sentiment

Some studies showed that there was an obvious correlation between the information of public opinion related to the community and the epidemic’s development. The headquarters also used the apps regarding community service to collect and analyze the residents’ needs. From theory to practice, many pieces of evidence showed that it was feasible to research the epidemic situation and the changing trend of community service through a network of public opinion information [13].

The results of the co-word analysis showed that residents had different concerns and needs in different stages of the epidemic (see Figure 5, Figure 6 and Figure 7). During the threshold period of the epidemic, residents were most concerned about the current situation of the epidemic, the prevention and control work of the community, and the community material support for residents’ lives. Most of them focused on the demand for life service support of supermarkets and “group buying,” and the services for screening patients with fever and community disinfection, etc. In the low-incidence period, there was not much difference compared with the previous period. Residents still paid more attention to the epidemic and the community prevention and control work, and their demands for livelihood and live support remained their major concern, while their demands for screening and disinfection services decreased. As the epidemic entered a normal period, residents paid less attention to the epidemic and the prevention and control work than before, and thanks to excellent achievements regarding the anti-epidemic work and effective measures taken by communities to ensure the livelihoods of residents, their attention and demands regarding all aspects stabilized.

Public opinion was consistent with the content of the actual service, which showed that the government and the community could give feedback to residents’ demands as soon as possible. The frequency of words, such as epidemic prevention and control, from the threshold period to the normal period decreased, while the frequency of words such as livelihood and life increased, indicating that community services were expected to shift from prevention and control to comprehensive services.

## 4. Discussion

### 4.1. Unreasonable Staff Assignments and the Structure of Community Services in the Early Stage of the Epidemic

At the beginning of the epidemic, the Wuhan community was faced with a lack of workers. Furthermore, a lack of professional health knowledge and inadequate protective measures cramped health-related work, such as the registration and transportation of patients, which greatly affected the implementation efficiency of epidemic prevention policies. These problems may have had something to do with the insufficient participation of community hospitals in the epidemic prevention and the control process since they failed to fully play the role of community medical practitioners in grass-roots prevention and control.

### 4.2. Community Services Optimized the Supply and Matched the Demand after the Adjustment of the Government Policy

Community services are important when fighting against an epidemic and maintaining people’s life quality, where the focus of services changed with the development of the epidemic and the needs of residents [14]. The results of the co-word analysis in this study showed that when the government adopted a policy of supplementing and refining services, the content of community objective services gradually transformed from epidemic prevention and control to comprehensive services. The transformation matched the change of the demand for community service and was consistent with the trend of the residents’ subjective expectations for community services during the epidemic. At the same time, with people having more chances to get medical help, the mortality rate was continuously decreasing, which indicated that the optimization of community service had a positive impact on epidemic prevention

### 4.3. As a Part of Urban Governance, Community Construction in China Needs to Strengthen Its Infrastructure

In China, the community has long faced a law dilemma. Characterized as the neighborhood committee, the community is burdened with a lot of administrative work but its legal status has been a gray area. A community is basically defined by the constitution as an urban grass-roots mass autonomous organization in China. It plays an extremely important role in urban communities. During the epidemic, the lower law was insufficient to cover the organic functions that communities performed, such as the mechanisms of the community and community medical institutions. In order to fully provide a public health function, the community needed legal support. Furthermore, community medical institutions should be included in the main body of community governance to assist in the epidemic prevention and the health maintenance work of the residents in the normal period.

### 4.4. Limitations of the Study

When using the DEA model to analyze the efficiency of community services in different stages, we only considered the number of community workers without analyzing the community workers capability and professional literacycapability. Furthermore, the number of community residents that came from the official community website was static data and could not reflect the changes in the community population in different periods. As an independent evaluation, the DEA’s results could not identify whether the community service system had reached an optimal state since the method did not take the input–output lag and sustainable characteristics into consideration.

Young people are the main population that expresses their personal opinions through the Internet. It is difficult to get the needs and opinions from the elderly and the infants via network monitoring and analysis. The data of the community services released by official channels and authoritative media may be slightly different from the actual services of various communities. The semantic analysis of words and sentences in the network text is easily influenced by personal subjectivity, and there is no set of fixed standards.

## 5. Conclusions

### 5.1. The Improvement of Community Services Will Play an Important Role in Epidemic Prevention and Urban Governance

During the epidemic, the Wuhan community was a typical and complex example of a health emergency community. Its practical elements included the community streets as autonomous organizations, primary medical institutions, and some commercial property organizations. Its responsibilities had various public attributes, such as public health, emergency support, and life services. The specific responsibilities included participating in the comprehensive management of sanitation; improving the public health environment in the community; assisting in the screening, transit, and treatment of patients; providing life service for non-COVID-19 patients and ordinary residents in the middle and late period of the anti-epidemic work. The community not only served as the undertaking organization of the public health emergency system but also fulfilled the typical urban governance role obligations. Their work enabled the epidemic prevention information to be released to the community in a timely manner, help patients with difficulties to effectively connect to an appropriate hospital, and allow for residents who are afraid to leave their homes for the risk of infection get help in life, which is of great significance to the prevention and control of the epidemic, the stability of order, and the recovery of the city. Some mature working modes and reasonable service contents will provide good methods for the whole city to deal with unexpected public health events and normal governance.

In the middle and late stages of the epidemic, the content of the community services gradually transformed from epidemic prevention and control to comprehensive services. The “residential economy,” which was magnified by isolation, significantly increased its dependence on community commerce, showing great potential. The “one-kilometer life circle” community business development space was greatly enhanced. In addition to the anti-epidemic measures, the complex and in-depth community economic activities were also constantly unfolding. The community paid close attention to the relevant policies of the government, examined the policies benefiting the people, provided answers and suggestions to the businesses according to the multi-dimensional problems with free consultation, continuously released the capital bonus, reduced the enterprise pressure, strengthened the business circle linkage, and activated the entity power framework platform. This established a commercial linkage mechanism to identify the needs of the people and the services that enterprises could provide during the epidemic period, connected each information transmission and resource dock, realized mutual benefits and win-win situations with strictly implemented measures, developed a safety net for epidemic prevention and control within the community, created a good business environment, and carried out the implementation of epidemic prevention and control measures, such as environmental sanitation, while correcting the deviation in time and ordering relevant subjects to check for errors and correct the measures used immediately.

As an economic organization with a unique social and economic development, the community had a unique microeconomic foundation regarding how to play the city’s function and promote economic growth. Doing a good job in community commerce is to meet the needs of the people’s livelihood. After the baptism of the epidemic situation, with the change of the consumption demand of the citizens, the community commerce aimed to find the right combination point and deeply investigated the market trend to understand the community concept and community service consciousness. After that, it repeatedly optimized and enriched the business planning, stimulated and enhanced the market vitality of the new area in the post epidemic era, met the needs of residents, expanded the new scale of community commerce, built and invested, and strived to create a model of community commerce under the new normal. In combination with the rigid demands of residents for community commerce, this new normal should also bridge the consideration of artistic, cultural, and professional services of community commerce; encourage more new consumption; upgrade consumption formats, such as healthcare, that are generated in the process of epidemic prevention and control. More online and offline services and scenes will continue to be integrated to boost the development of more enterprises and wake up the city’s stronger vitality.

### 5.2. The Improvement of Community Public Health Capacity Plays a Fundamental Role in Responding to Public Health Emergencies

The experience of community epidemic prevention and control work in Wuhan suggests that it was necessary to continuously optimize the deployment of community human resources, especially the number and capacity of health technicians in the community, for which their professional performance mad them better guide administrators and business people in community service, thus improving the coping efficiency. Selecting and rewarding excellent community doctors for guidance and consultation in community daily public health work should not be limited to the care of common and chronic diseases. Detailed community public health emergency guidelines should be issued and community hospital staff skills training should be organized in response to public health emergencies. These initiatives will be useful for enhancing the capacity of the community regarding public health services.

In the application of epidemic situation tracking, China allowed for the full mobilization ability of grassroots communities, implemented carpet tracking and grid management, implemented prevention and control measures to households and people, and provided mass and stable prevention and control. Through the establishment and improvement of epidemic prevention and control mechanisms and the grid work system, China built a full-time and part-time work team to fully support medical staff and family doctors in primary medical and health institutions. With the joint efforts of the student team, the big data platform was integrated to improve the sensitivity and refinement of tracking. To ensure the tracking of people, registration, community management, door-to-door observation, the community strengthened the services and observation of key groups, in combination with property management, conforming to the inquiry, temperature measurements, registration, and other processes; furthermore, it carried out effective transportation with superior medical institutions to achieve the epidemic prevention and control goal of the “prevention of import, spread, and output,” and providing convenient and quick registration and filing channels for the masses by means of information technology. At the same time, to comprehensively improve the effectiveness of the epidemic prevention and control and provide effective support throughout the whole region, a person leaving or arriving responded to a questionnaire by scanning a two-dimensional code through WeChat, filling in the information, such as name, ID card number, contact number, residential address, work unit and time of going out (return), reason for going out, destination (origin), return time, and body temperature. After the community personnel clicked the report button to complete the information registration and filing, the district grid management service center made a daily statistics summary flow and organized and reported the information and data submitted by the personnel to the superior department.

### 5.3. The Chinese Community Governance Model Is Unique and Shared

The self-government management mode of the American community is mainly realized by non-profit organizations and volunteers, which are used as leverage to manage and serve the community, forming a partnership with the government. The government guides and supervises the work of social intermediary organizations and volunteers, which also plays an important role in maintaining social stability. The community governance in Singapore is dominated by the government. With the combination of autonomy, the government provides methods to guide the community to carry out autonomous management, guides social organizations and enterprises to actively participate in community construction, and jointly establishes and improves the community service ecological network by providing corresponding assistance for different groups and expanding the full space for the development of community organizations.

During the epidemic period, the Chinese community governance was unique, though it is still worth learning from. During the epidemic period, the Wuhan municipal government used the market mechanism to introduce commercial services to enrich the human resources of community services and strengthened the anti-epidemic force, developed and applied a community service docking platform through an innovative service model to respond to residents’ dynamic demands, realized the service contents and comprehensively arranged personnel around a fixed-point, improved the use efficiency of funds, and promoted the development of related industries. In fact, through the practice of community service optimization during the epidemic period, scientific urban governance was promoted in the normal period. According to local conditions, other countries can also use mature commercial service supply channels to enrich anti-epidemic forces by means of market purchases, encouraging enterprises to participate, developing community service docking platforms and apps, responding to the needs of residents, setting up service contents and arranging personnel, and exploring various methods of urban governance during an epidemic period.

## Figures and Tables

**Figure 1 ijerph-17-07633-f001:**
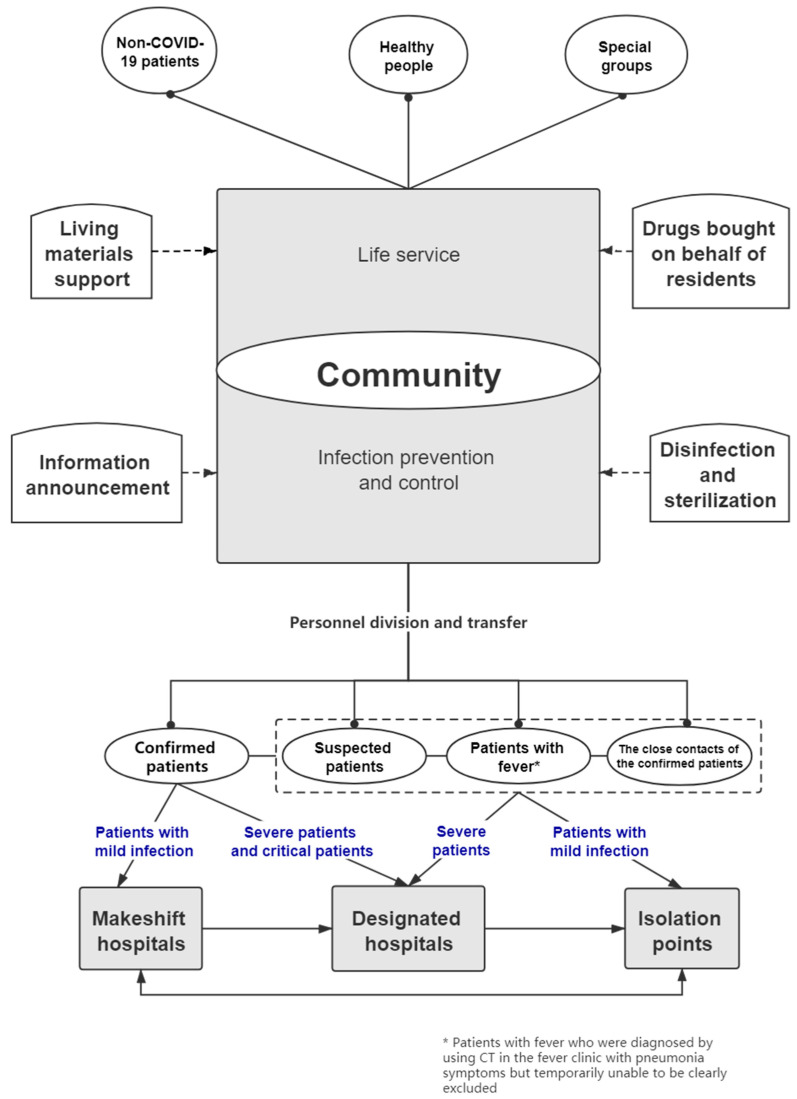
Schematic diagram of the streaming process from the community to different types of hospitals of different types of patients.

**Figure 2 ijerph-17-07633-f002:**
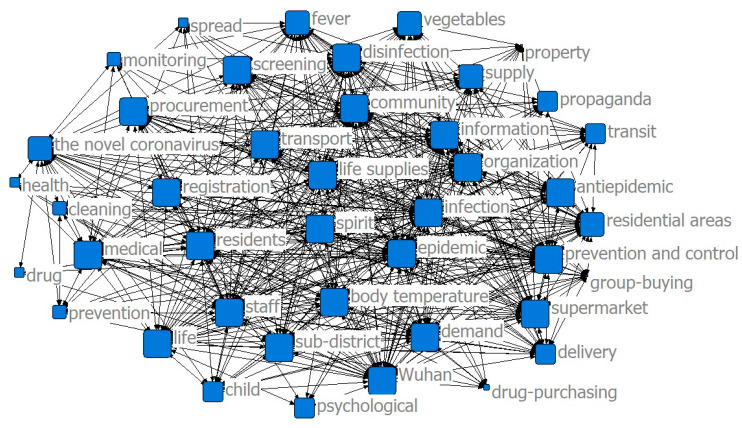
Community services in the threshold period.

**Figure 3 ijerph-17-07633-f003:**
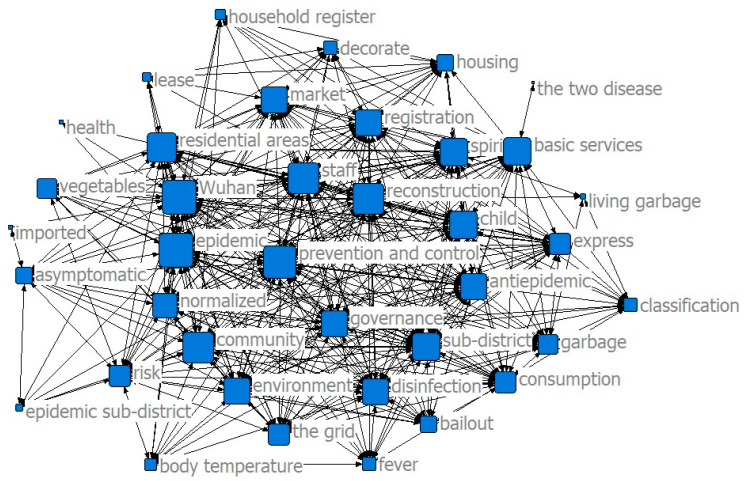
Community services in the low-incidence period.

**Figure 4 ijerph-17-07633-f004:**
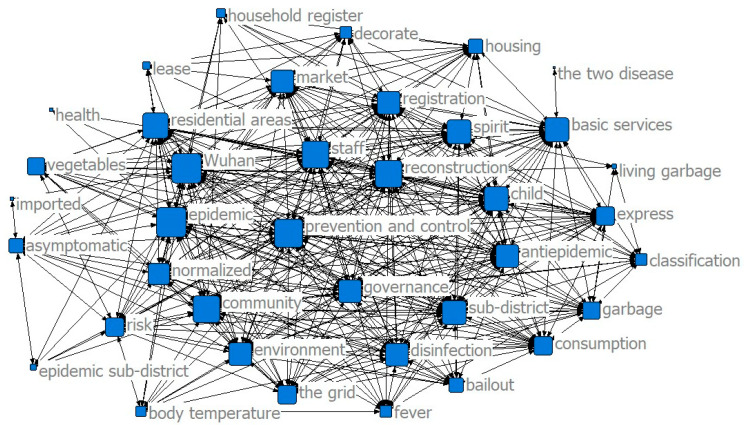
Community service content in the normal period.

**Figure 5 ijerph-17-07633-f005:**
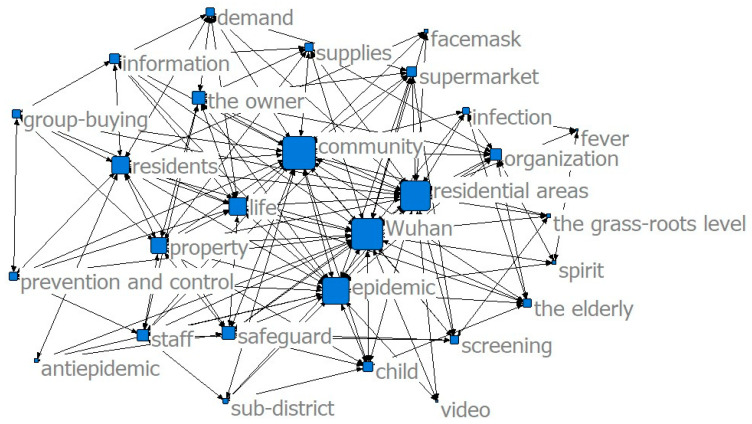
Community service demands in the threshold period.

**Figure 6 ijerph-17-07633-f006:**
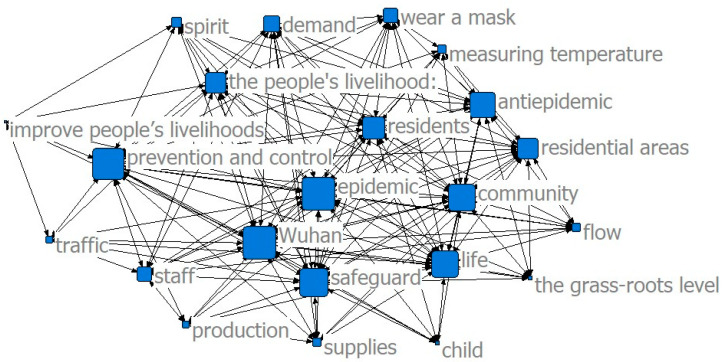
Community service demand in the low-incidence period.

**Figure 7 ijerph-17-07633-f007:**
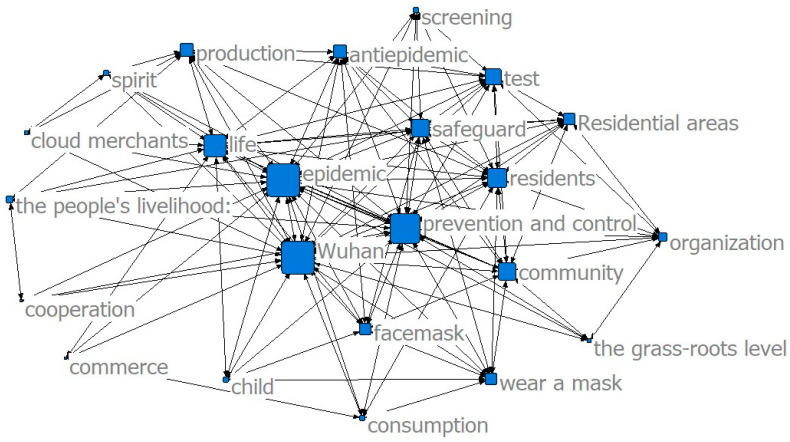
Community service demands in the normal period.

**Table 1 ijerph-17-07633-t001:** Indicators of the data envelopment analysis (DEA) model.

Classification	Indicator’s Name
Input	Number of community workers
Output	Number of community residents
	Number of confirmed patients in the commmunity

**Table 2 ijerph-17-07633-t002:** Results of the DEA efficiency evaluations about communities in Wuhan in the early stages.

Community	te	crste	vrste	scale	rts
Caidian District	0.213	0.213	0.228	0.932	irs
Donghu Ecotourism Scenic Area	0.633	0.633	1.000	0.633	irs
New Technology Development Zone of Donghu	0.733	0.733	0.764	0.960	irs
East Donghu District	0.523	0.523	0.549	0.953	irs
Hanyang District	0.821	0.821	0.842	0.975	irs
Hongshan District	0.782	0.782	0.784	0.997	irs
Huangpi District	0.225	0.225	0.228	0.990	irs
Jiangan District	0.882	0.882	0.886	0.995	irs
Jianghan District	0.890	0.890	0.906	0.983	irs
Jiangxia District	0.293	0.293	0.302	0.968	irs
Qiaokou District	1.000	1.000	1.000	1.000	-
Qingshan District	0.600	0.600	0.630	0.952	irs
Wuchang District	1.000	1.000	1.000	1.000	-
Wuhan Economic and Technological Development District	0.569	0.569	0.615	0.924	irs
Xingzhou District	0.203	0.203	0.207	0.984	irs

Notes: te is the technology efficiency; crste is the overall efficiency; vrste is the true technical efficiency; scale is the scale efficiency; rts is the return to scale; irs signifies an increase; - signifies a constant value.

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
