# Peer review of "Exploration of the Evaluation and Optimization of Community Epidemic Prevention in Wuhan Based on a DEA Model"

_ijerph, 2020, doi:10.3390/ijerph17207633_

Round 1

Reviewer 1 Report

The paper requires substantial editing of language and style.

This paper is interesting as it characterizes the response to COVID-19 by evaluating the level of community services during the outbreak at the different stages from the start to peak and recovery from the outbreak.

Line 53 It would be better to change the sentence to Other countries have a mixture of private and government services for community infrastructure.

In the discussion

The governance model of China should be characterized with governance models from other countries such as the U.S where complete lockdowns are not feasible and discuss the implementations for public health.

The use of covid-19 tracing apps in China should be discussed and how these apps were able to control the outbreak.

Author Response

1.We have increased the discussion about the Implementations for public health in other countries and how these apps were able to control the outbreak.

Reviewer 2 Report

The manuscript deals with an interesting issue in Public health.

However, several issues must be faced with in order to be suitable for publication.

Major issues

  • in the title the authors reported they performed a systematic review, but in the text there is no explanation on the methods they used for systematically review the issue. Please clarify
  • The aim of the study must be clearly reported.
  • the introduction section must be focused mainly on the objective of the study.
  • the methods section lacks of clarity concerning the way in which efficiency was measured. Please give more details on how the method was used
  • Co-word analysis and Social network analysis are only mentioned. the authors must clarify how they used these analyses in their study.
  • in the description of the co-word analysis there is a lack of clarity of the comparisons in the three periods. Are there any additional way in presenting data (a table?)?
  • a paragraph on the limitations of the study is required. the authors need to specify what bias could have been present and how they tried to avoid. What is the internal validity? and what about the external validity?

Minor issues:

  • the abstract is almost uninformative in the results section. Gve more details, especially considering the efficiency figures.
  • since the authors presented the manuscipt as a systematic review, the PRISMA checklist must be presented

Author Response

We wrote a new paragraph to describe the purpose of the study, and explained in detail how the DEA and co-word analysis methods were used in the methods section. Finally, we added a paragraph about the limitations of this study.

Reviewer 3 Report

This is a study that helps evaluating the community epidemic prevention in Wuhan using DEA model.  I have the following comments/suggestions:

1. I found a lot of misspelling throughout the manuscript.  Please look at line 60, 85, 228, 253, 270, for example.  Please double check the whole manuscript for other misspelling.

2. Line 12, 14: please indicate the year

2. Line 111: please give the full name of "CCD"

3. Figure 2-7: to make it easier for readers, please provide brief instruction how to interpret it

4. Please comment on the weakness/strength of the methods using in this study.

Author Response

We carefully check the spelling of the article and explain in detail how DEA and co-word analysis are used in the methods section. Finally, we add a paragraph about the limitations of this study.

Reviewer 4 Report

Zhang and colleagues systemically reported the evaluation and optimization of community epidemic prevention in Wuhan using Data Envelopment Analysis (DEA) model. The authors collected community manpower data by semi-structured interview and evaluated the service supply and demand balance during epidemic with co-word analysis method. They identified that the community services and material supply were in shortage during the epidemic. Government intervention improved the demanded service and shortage of supply with measures of setting up targeted community service projects, coordinating volunteers, and wisely using information platform. The emphasis of community service was gradually transformed from epidemic prevention and control to integrated service. The authors concluded that the government should improve the policy on community service in epidemic control and economic recovery after the epidemic. However, there are some issues could be addressed.

Major comments,

  1. Overall the manuscript is poorly written. The logic, organization, and presentation of the data are also major concerns. The description of DEA in the method part should move to the introduction part. When describing DEA model in Method section, the authors should provide more detailed information on how to perform DEA and the data analysis. In addition, in manuscript writing, it’s better to summarize the results with one or two sentences at the end of each paragraph. The detailed conclusion could be moved to Discussion section.
  2. The authors presented the data of the DEA efficiency value in table 2. However, there is no threshold of this value presented in the paper. The value indicated should be addressed.
  3. The discussion part should be more focused on the data and facts presented in the results. Detailed discussion of community supplement in the control of pandemic and economic recovery after pandemic should be noted or cited.
  4. The English in the text is poor and hard to be understood. There are numerous grammatical and typographical errors throughout the text, should be corrected. For example, in table 1, the word of “Inpute” should be input. In line 139, the content of “(2)” should be deleted.

Author Response

We moved the description of the method section to the introduction section. In the method, more detailed data envelopment analysis and data analysis information are carried out.

A detailed discussion of communities in terms of pandemic control and post-pandemic economic recovery is complemented.

We rechecked the grammar and spelling mistakes.

Round 2

Reviewer 1 Report

The authors have improved the paper by addressing the reviewers comments.

Reviewer 2 Report

The manuscript can be accepted for the publication

Reviewer 4 Report

I do not have further concerns.